# Global Self-Esteem, Physical Activity, and Body Composition Changes Following a 12-Week Dietary and Physical Activity Intervention in Older Women

**DOI:** 10.3390/ijerph192013220

**Published:** 2022-10-14

**Authors:** Mateusz Grajek, Agnieszka Gdańska, Karolina Krupa-Kotara, Joanna Głogowska-Ligus, Joanna Kobza

**Affiliations:** 1Department of Public Health, Department of Public Health Policy, Faculty of Health Sciences in Bytom, Medical University of Silesia, 40-055 Katowice, Poland; 2Department of Epidemiology, Faculty of Health Sciences in Bytom, Medical University of Silesia, 40-055 Katowice, Poland

**Keywords:** global self-esteem, older women, diet-reducing, physical activity, COVID-19

## Abstract

Older adults show lower physical activity. These changes altogether promote the development of overweight, obesity, and other chronic diseases. These factors substantially influence the quality of life and self-esteem of older adults. This phenomenon is especially visible after the lockdown caused by the COVID-19 pandemic. Objectives: Our study aimed to evaluate the effect of a 12-week reductive diet and a 12-week physical activity plan for older adults on the global self-esteem of lifestyle in 60–70-year-old women. Materials and methods: Our participants were 600 women with increased body mass (BMI > 25 kg/m^2^) aged 60–70 years. After the initial evaluation, the participants were randomly divided into three groups: CG—control group (*n* = 200); DI—dietary group (*n* = 200) that committed to a 12-week reductive diet; PA—physical activity group (*n* = 200) that committed to a 12-week physical activity plan. The global self-esteem score (using the SES Rosenberg scale) and the anthropometric measurements were collected before and after the 12-week study. In the statistical analysis of data, the significance level was assumed to be 0.05. Results: The global self-esteem score for all groups before the study started was 30–31 points, which corresponded to average self-esteem. After a 12-week dietary or physical activity intervention, the score in the DI group was 33, which corresponded with high self-esteem. In the CG group, the self-esteem score remained unchanged (30 points). The average body mass loss was 0.5 kg/m^2^ for CG, 1.92 kg/m^2^ for DI, and 1.10 kg/m^2^ for the PA group. The average waist–hip ratio (WHR) change for CG, DI, and PA was 1 cm, 1 cm, and 2 cm, respectively. A decrease in body mass and body composition indicators (BMI and WHR) corresponded to participants’ global self-esteem increase (*p* < 0.05); the greater the decrease noted for BMI and WHR, the greater the global self-esteem score that was achieved. In the CG group, a negative correlation between global self-esteem and BMI value (*p* < 0.05) was observed. Conclusions: A 12-week reductive diet and a 12-week regular physical activity plan lowered participants’ body mass. Adipose tissue content was reflected by decreased BMI and WHR indicators of participants from the DI and PA groups and was accompanied by higher global self-esteem scores.

## 1. Introduction

COVID-19 has spread around the world, causing intense fear of infection, which can lead to permanent anxiety. At the same time, social quarantine and difficult access to medical services may result in the deterioration of mental and physical health (including weight gain) [1]. Given the escalation in many cases, humankind is facing countless psychological problems, from those related to taking precautions and maintaining safety to those caused by separation [2]. From the somatic point of view, the biggest problem associated with isolation was the lack of possibility of direct contact with medical professionals, and thus the aggravation of many chronic (coexisting) diseases. In many older adults, after months of lockdown, increased body weight and low levels of physical activity were observed [3]. Researchers suggest that the high self-esteem associated with confidence and the body acts as a shield against mental health risks [4]. To maintain self-assessment at an appropriate level, one should take care of the somatic sphere (proper diet and physical activity) [4].

Self-esteem is a relatively stable attitude towards the self and is affected by many factors including cultural, ethnic, and social factors [5,6]. Previous studies on self-esteem have indicated that self-esteem is also affected by social status, social relations, professional situations, and lifestyle [7,8]. Currently, several areas influence people’s self-esteem. These areas form the so-called global self-assessment (GSE), which consists of such aspects as assessment of one’s competence, self-assessment of the quality of close and intimate interactions with other people, self-assessment of popularity among friends and social attractiveness, assessment of one’s leadership skills, self-control, and moral self-acceptance including physical attractiveness and vitality [6,9].

It is estimated that during the next 50 years, the demographic trend will change, and there will be a considerable increase in the number of older people in society. This change will be caused by an increase in life expectancy observed in recent years [10]. Life expectancy will not always be linked to quality of life. Older people frequently experience chronic diseases, and their functional and psychological well-being is deteriorating [11,12]. With age, the aging process intensifies, which increases many chronic diseases [12,13]. The characteristic elements of aging also include the reduction in muscle and bone mass with a simultaneous increase in body fat. These factors affect posture defects and cause a change in body shape [14]. In addition, changes in the gastrointestinal tract (e.g., impairment of liver function, slowing down of intestinal peristalsis, and a decrease in digestive enzyme activity) may contribute to the slowing down of metabolism and abnormal dietary habits that cause overweightness and obesity [12,15,16]. Scientific reports that are based on the evaluation of health indicators in older adults have suggested that future care needs for older adults will require much higher medical and long-term care costs than anticipated [13,16]. Unhealthy behaviors are prevalent among older adults, yet poor health does not necessarily dominate the older population. Most older adults’ health problems are caused by chronic diseases that can be prevented or delayed by changing lifestyles and introducing healthy habits [17]. Physical activity and good nutrition provide great benefits to health and well-being in old age and can positively affect mental health [14,17]. Therefore, physical and mental health and high self-esteem stand out among the factors that contribute to successful aging [10].

Thus, this study aimed to assess the effect of a 12-week reduction diet and 12-week physical activity plan on the weight reduction and global self-esteem of 60–70-year-old women after the COVID-19 lockdown.

## 2. Materials and Methods

### 2.1. Study Organization and Eligibility Criteria

The study group consisted of 600 overweight or obese 60–70-year-old women who participated in classes at 3rd century universities in Silesia (Poland). The study was conducted after the lockdown (June–August 2020) caused by the COVID-19 pandemic.

The exclusion criteria in the study were ages <60 and >70 years, diseases requiring a special diet, and factors that may limit daily physical activity such as locomotive organ dysfunction, cardiovascular diseases, diabetes, and hypertension. Women showing elevated BMI values requiring dietary intervention of >25 kg/m^2^ were eligible for the study. Before examination, the women had a medical consultation to exclude contraindications to participate in the study. The consultation was conducted by a general practitioner and a dietitian. The doctor conducted a basic medical history check combined with a baseline examination, and the dietitian was responsible for conducting a nutritional history check and body composition analysis.

The target group for the survey was 870 people, but taking into account the eligibility criteria and minimum sample size, 600 people finally qualified for the survey. The assignment of individuals to subgroups was based on random sampling. Participants drew numbers (123, 213, and 312) from an urn, the meaning of which was not understood by them, but coded only to the survey organizers. Participation in the study was voluntary. The participants were informed about the purpose and course of the study, and about the possibility of resigning from participation in the research at any stage. The project was presented to the Bioethics Committee, which accepted it for implementation.

The women were divided into three equal groups (200 people each):DI—diet intervention group—a group with a 12-week intervention consisted of the introduction of an individualized reduction diet, and care of a dietitian was provided;PA—physical activity group—a group with a 12-week physical activity program for seniors, tailored to individual needs, was conducted by a fitness trainer qualified in senior activity;CG—control group—without any intervention.

The minimum group size was estimated based on a sample size formula calculated from the average size of the overweight population for the Silesian population (Poland) [18]. There were no statistical correlations between the groups in their structure in terms of socio-demographic data. All the participants that qualified for the study participated until the very end.

### 2.2. Study Procedure, Research Tool, and Interpretation

During the study, the daily level of physical activity was reported by participants and monitored by a dietitian at visits. The level of GSE (the results of ratings on multiple dimensions, including self-assessment of appearance, intelligence, and self-confidence, as measured by Rosenberg’s SES scale) before and after the 12-week intervention was assessed. Anthropometric measurements were obtained (body height, body weight, waist circumference, hip circumference, and gluteal muscle protrusion), and the following body mass indices were determined: BF (kg)—body fat; LBM (kg)—lean body mass. Their percentages (%BF and %LBM) were measured by the electrical bioimpedance method (Tanita TBF-300M); BMI (Body Mass Index) and WHR (waist-to-hip-ratio) had the following standards: BMI 17–18.49 kg/m^2^—underweight; 24.9–29.9 kg/m^2^—overweight; >30—obesity (WHO 2018) and WHR ≥ 0.80 (WHO 2011).

Pedometers from Yamax Inc. (Kumamoto, Japan) were used as an objective tool to assess physical activity. For physical activity assessed with pedometers, the following standards were adopted according to Oliveira [19]: <5000 steps/day—sedentary lifestyle; 5000–7499 steps/day—low physical activity; 7500–9999 steps/day—moderate physical activity; >10,000 steps/day—desired physical activity. The surveyed women were instructed on how to use the pedometer. The tool was placed on the lower limb [20]. The pedometer estimated the number of steps per week. The pedometer was removed only for bathing and sleeping.

#### 2.2.1. Diet Intervention

In the group with diet intervention (DI), the analysis of the current diet was performed based on a questionnaire using a five-day dietary history check. Meal sizes (g) were estimated based on an illustrated photo album of products and dishes. The analysis of dietary composition was developed using the DietPerfekt and Dieta 6 dietary programs (Poland) based on actual nutritional norms [21]. The proportions of nutrients were calculated according to the amended dietary standards for the Polish population [22]. The surveyed women kept daily food consumption diaries, which were designed to minimize dietary mistakes and were checked by a dietitian during weekly obligatory check-up visits. Daily energy consumption in the diet was reduced by 20% of total metabolism. The amount of mono- and polyunsaturated fatty acids and dietary fibers (>30 g/day) was increased, while cholesterol (<300 mg/day), sodium, and added sugars were reduced. The diet was introduced and controlled for 12 weeks.

#### 2.2.2. Physical Activity Intervention

In the group with physical activity intervention (PA), physical activity was introduced 3 times a week in the form of individualized gymnastics for seniors. The physical effort was adjusted to the capabilities and health of the participants. The participants were informed about the necessity of performing exercises according to their abilities. The proposed physical activity was determined based on current WHO guidelines. Each one-time training lasted 45 min and included aerobic effort, resistance exercises, stretching exercises, and exercises aimed at improving balance and preventing falls (PAL_max_ = 1.6).

All measurements were checked every 4 weeks from the start of the study together with the body composition analysis.

In the control group (CG), a one-time meeting with a dietitian and a personal trainer was conducted. The meeting was divided into two blocks: educational activities with a dietitian (including anthropometric measurements and body composition analysis) and activities with a personal trainer.

The GSE of the surveyed women was determined using a questionnaire containing the SES scale developed by Rosenberg [23], which was the Polish adaptation of Łaguna [24]. All participants were asked to complete the questionnaire immediately before and after the introduction of the selected intervention. The Rosenberg scale is one of the tools used to perform analysis to determine the relation to self. The sum of points obtained during the completion of the questionnaire is an indicator of the level of self-assessment. The range of points possible to obtain is from 10 to 40; specifically, 10–27 points—low self-esteem; 28–32 points—medium self-esteem; 33–40 points—high self-esteem.

### 2.3. Statistical Analysis

The data were reported as the mean ± standard deviation (SD). A two-way repeated measures ANOVA with three groups (CG—control group; DI—diet intervention; PA—physical activity) and two measurements before and after the 12-week intervention was applied. When appropriate, the Tukey post hoc comparison was used. Relationships between variables were evaluated using Pearson’s correlation coefficient. The level of significance was set at *p* < 0.05. Statistical analyses were performed using Statistica 13.3 (Statsoft, Tulsa, OK, USA).

## 3. Results

The study included 600 women between the ages of 60 and 70. The average age was 64 + 3 years. All women showed an elevated BMI indicating at least overweight. In addition, all women were classified in terms of material status as having a typical income for the Polish population (PLN 3200 per person in the family). A total of 80% of the subjects had secondary education, 12% had higher education, and 8% had vocational education.

It was observed that both the 12-week change in diet (introduction of a reduced diet) and the introduction of physical activity had a positive effect on the improvement of anthropometric (BMI and WHR) and body composition indexes (%BF and %LBM) in groups with a reduction diet (DI) and increased physical activity (PA) compared to the control group (CG). In the DI and PA groups, the average weight loss was 4.91 kg and 2.76 kg, respectively, while for CG, it was 1.33 kg. For BMI, the decrease for CG, DI, and PA groups was as follows: 0.52 kg/m^2^; 1.92 kg/m^2^; 1.10 kg/m^2^. The following results were obtained for WHR in the same groups: 1 cm; 1 cm; 2 cm. The difference in body composition indices was in the following groups: %BF (1.34%; 3.94%; 3.92%) and %LMB (−1.34%; −3.94%; −3.92%). Changes between groups within the given interventions were shown to be statistically dependent. The greatest changes were shown in the DI group (*p* < 0.05)—Table 1.

The increase in the level of physical activity in the groups was as follows: CG—increase of about 4%; DI—3%; PA—10%. Despite the interventions introduced, they remained at a low level (*p* < 0.05)—Table 2.

After the interventions, a significant increase in GSE expressed in the Rosenberg scale was also observed. The most significant increase in self-esteem was recorded in the DI group—from average (31 points) to high self-esteem (33 points). In the PA group, the result still oscillated within the limits of the average self-esteem (about 31 points). In the CG group, after 12 weeks, there were no significant changes in GSE value. The obtained results suggest that weight reduction achieved by changing diet or physical activity has a positive effect on improving GSE. The results are statistically significant (*p* < 0.05)—Table 3.

## 4. Discussion

Regular physical activity in old age improves physical, mental, and social functions and reduces the effects of chronic diseases and civilization. A low level of physical activity promotes the development of overweight and obesity and negatively affects the process of changing body structure and posture [25]. Currently, the study attempts to assess the effect of regular physical activity on the level of GSE and life satisfaction in different age groups and populations [26]. Many previous studies have shown that an increase in physical activity has a positive effect on self-esteem, while a sedentary lifestyle is associated with a decrease in physical activity [27,28,29,30]. A study by Masana et al. [31] has shown that older people with low self-esteem or anxiety disorders are considerably more likely to lead an unhealthy lifestyle, which is characterized by low levels of physical activity or the use of stimulants such as smoking. Similar results were obtained by McAuley and Blissmer [25], who analyzed the effect of regular physical activity in old age on performance indicators, body perception, and self-esteem. These authors have shown that undertaking regular physical activity is associated with a higher level of self-esteem, and this increase in self-esteem is due to an improved perception of the attractiveness of the body.

In the current study, interventions, both dietary- and physical-activity-based, were shown to improve baseline anthropometric indices, such as BMI and WHR, as well as body composition indices (%BF and %LMB) in the study groups. The best effect of the introduced interventions is seen in the reduction diet group. In this group, the differences between BMI, WHR, %BF, and %LMB before and immediately after the intervention are the greatest. The introduction of dietary or physical activity interventions and the consequent weight loss has a clear effect on raising the subjects’ GSE. The performed study showed that the introduction of a diet with reduced energy (the daily energy consumption in the diet was reduced by 20% of total metabolism) about demand resulted in weight reduction and was the most beneficial of all interventions; in the DI group, the body weight was reduced by 6.57%; the body weight was reduced by 3.67% in the PA group; and by 1.81% in CG. In the study group, diet also considerably increased the level of GSE (33.93 ± 4.62). There was observed a statistically considerable increase in the level of GSE of seniors after 12 weeks of regular physical activity, although the GSE value was determined to be average (i.e., 31.24 ± 3.39). Despite considerable changes after 12 weeks of intervention, the average BMI value in this group was still in the overweight range.

Furthermore, most of the previously performed studies on the relationship between lifestyle and self-esteem levels are based on the analysis of individual health behaviors (e.g., only physical activity or diet) [27]. These studies do not provide clear conclusions regarding whether the quality of diet, physical activity, and sedentary lifestyle are independent factors that affect self-esteem [32]. In addition, other studies on self-assessment showed that weight reduction had a positive effect on the sense of attractiveness and an improved perception of the body. In addition, these factors considerably affect changes in self-esteem in obese people. The study by Masana et al. [31] has suggested that eating habits and unhealthy lifestyles in older people are strongly correlated with low self-esteem, anxiety disorders, and depression. These studies have also shown that older people with low self-esteem or anxiety disorders are much more likely to lead unhealthy lifestyles. In addition, these relationships were conditioned by family status and gender; problems of abnormal eating habits and lowering of mood and depression were more common in older women. Unfortunately, there is still a lack of clear research in the literature on the effect of dietary change and weight reduction in seniors on the level of GSE [33]. Additionally, the study by Bherer et al. [34] and Weinberger [35] has shown that weight loss in overweight and obese individuals must be above 10% of their initial body weight for this to have a positive effect on their body acceptance and GSE. A positive difference in the level of body fat was also observed; however, the studied women still had an average level of GSE before and after the intervention, which suggested that their results were not satisfactory in the context of changing their body image.

The introduction of a 12-week intervention consisting of a reduced diet or regular physical activity in the examined seniors improved body weight, anthropometric indicators, and body composition. In both cases, there was an increase in GSE rating [29]. When a diet was introduced, the level of global self-assessment changed from medium to high. The obtained results suggest that the key role in the change in GSE may be played by the change in body weight and the level of adipose tissue associated with the change in figure [30]. Body image and self-esteem have already been empirically linked; however, the relationship between them in older people requires further research. The results obtained in this self-assessment study indicate that there is a link between global self-assessment and self-esteem in older adults and suggest that psychological interventions in the target population of seniors should be planned [36].

## 5. Strengths and Limitations

Undoubtedly, a strong point of the conducted study is the large study group, which constitutes a representative group, as far as research conducted in these groups is concerned. The use of appropriately selected tools and methods significantly reduced the risk of error. The survey was conducted using the face-to-face survey method, which avoided the common phenomenon of “bot/fake responders” in surveys that were mainly conducted online during the lockdown period. In addition, there is a high risk of bias in the results presented, as they were reported on an ongoing basis by the participants and properly evaluated and read off during the inspections, which took place three times at 4-week intervals.

Due to the nature of the group, the results may not be generalizable to other populations, such as the general healthy population or people with generally low self-esteem. Further research should include other research groups—inactive seniors (who do not participate in senior clubs or University of the Third Age classes), older people (over 70), or seniors with low self-esteem.

## 6. Conclusions

The interventions introduced in the study groups, both dietary and physical activity, improved basic anthropometric indicators such as BMI or WHR and body composition indicators (%BF and %LMB). The best effect of the introduced interventions is visible in the reduction diet group. In this group, the differences between BMI, WHR, %BF, and %LMB before and immediately after the intervention are the highest. The introduction of dietary- or physical-activity-related interventions, and consequently weight loss, has a clear impact on the level of global self-esteem in older women. The most significant increase in self-esteem is caused by the effects of dietary interventions. The control group has not seen any significant changes either in anthropometric indicators, body composition, or in global self-assessment results. This suggests that the results obtained in the study may be related to the general population.

## Figures and Tables

**Table 1 ijerph-19-13220-t001:** Anthropometric indicators—differences between groups (*n* = 600).

Variable	12-Week Intervention *			SS	MS	dF	F	*p*
	CG *n* = 200	DI *n* = 200	PA *n* = 200					
Body mass [kg/m^2^]	73.12 ± 8.64	74.71 ± 10.22	73.50 ± 12.69	272.20	272.20	2	67.45	0.001
	71.79 ± 8.45	69.80 ± 9.70	70.74 ± 11.65					
Body fat [kg]	26.96 ± 5.01	28.43 ± 7.26	28.35 ± 8.73	341.28	341.38	2	62.76	0.001
	25.55 ± 5.09	23.62 ± 4.70	24.50 ± 7.80					
Body fat [%]	36.69 ± 3.83	37.66 ± 4.80	38.03 ± 4.69	284.00	284.00	2	56.31	0.001
	35.35 ± 3.91	33.72 ± 3.54	34.11 ± 4.49					
Lean body mass [kg]	46.16 ± 3.83	46.28 ± 4.54	45.15 ± 5.09	3.90	3.90	2	0.91	0.001
	46.65 ± 3.91	46.18 ± 6.22	46.24 ± 5.05					
Lean body mass [%]	63.31 ± 3.83	62.34 ± 4.80	61.97 ± 4.69	284.00	284.00	2	56.31	0.002
	64.65 ± 3.91	66.28 ± 3.54	65.89 ± 4.49					
BMI	29.55 ± 2.70	29.33 ± 3.66	29.66 ± 4.89	41.80	41.80	2	63.57	0.001
	29.05 ± 2.89	27.41 ± 3.51	28.56 ± 4.56					
WHR	0.91 ± 0.07	0.84 ± 0.18	0.90 ± 0.06	0.0079	0.0079	2	29.56	0.003
	0.90 ± 0.02	0.83 ± 0.18	0.88 ± 0.05					

* The first line shows the values before the intervention, and the second line shows the values after the intervention. Legend: SS—sum of squares; MS—sampling variance; dF—degrees of freedom; F—statistical test; *p*—probability.

**Table 2 ijerph-19-13220-t002:** Level of physical activity—differences between groups (measurement from pedometers)—*n* = 600.

Variable	12-Weeks Intervention *			SS	MS	dF	F	*p*
	CG *n* = 200	DI *n* = 200	PA *n* = 200					
Pedometric measurement	5896.00 ± 2817.27	6240.08 ± 3431.3	6143.19 ± 3750.66	3.6	3.6	2	39.1	0.001
	6151.40 ± 2907.97	6435.53 ± 3431.29	6691.34 ± 2278.50					
Interpretation	low physical activity	low physical activity	low physical activity	-	-	-	-	-

* The first line shows the values before the intervention, and the second line shows the values after the intervention. Legend: SS—sum of squares; MS—sampling variance; dF—degrees of freedom; F—statistical test; *p*—probability.

**Table 3 ijerph-19-13220-t003:** Global self-assessment scores—differences between groups (Rosenberg scale)—*n* = 600.

Variable	12-Weeks Intervention *			SS	MS	dF	F	*p*
	CG *n* = 200	DI *n* = 200	PA *n* = 200					
Rosenberg test value	30.33 ± 2.92	31.55 ± 5.08	30.10 ± 3.63	15.40	7.70	2	7.17	0.002
	30.00 ± 4.63	32.93 ± 4.62	31.24 ± 3.39					

* The first line shows the values before the intervention, and the second line shows the values after the intervention. Legend: SS—sum of squares; MS—sampling variance; dF—degrees of freedom; F—statistical test; *p*—probability.

## Data Availability

The data presented in this study are available on request from the corresponding author.

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
