# Peer review of "Global Self-Esteem, Physical Activity, and Body Composition Changes Following a 12-Week Dietary and Physical Activity Intervention in Older Women"

_ijerph, 2022, doi:10.3390/ijerph192013220_

Round 1
Reviewer 1 Report
To the authors, please address the following points:
Thank you for the opportunity to review this manuscript. Overall, I feel there is insufficient detail regarding the study participants, the intervention descriptions and random sampling method. Also, overall, the quality of references needs to be improved. As such, revisions are required.
1) Background - Line 43 –“Researchers suggest ….” should end with reference to the Rossi study.
2) Background - Line 43 – Please clarify self-assessment of what? Once clarified, please correct all mentions of this throughout the manuscript.
3) Background – Line 55 – (second sentence of third paragraph) - Suggest that you update the World Health Organization reference from 2002 to reflect your sentence “…an increase in life expectancy observed in recent years”.
4) Materials and methods – study organization and eligibility criteria – first sentence (line 80) please make clear what you mean when you write “…participated in classes at the University of the 3rd century and active….” And clarify the setting of these classes, e.g., city.
5) Materials and methods – study organization and eligibility criteria – please clarify the eligibility criteria for recruitment of participants, including in criteria - overweight or obese.
6) Materials and methods – study organization and eligibility criteria Line 91 – suggest you clarify the random sampling method.
7) Please clarify how you determined your sample size?
8) Materials and methods – study organization and eligibility criteria line 86 – suggest you provide additional detail to support the sentence “Before the examination, the women had a medical consultation to exclude contraindications to participate in the study.” For example, setting, staff involved, content etc.
9) Materials and methods – study organization and eligibility criteria – line 96 – 108 is repetition of this section. Please remove.
10) Study procedure, research tool, and interpretation – Please carefully revise the reporting of the intervention descriptions and the references referred to in the reference list. For example, line 110 – In first sentence please make clear how and who received monitoring of physical activity levels. Also, in the second sentence, please clarify what you mean when you say “global self-assessment. In the 3rd sentence (line 112) please clarify what you mean when you say the “examined women”. Line 121 – please clarify if Oliveira is the correct source of evidence from which to set steps/day standards. In the description of the motor intervention, line 128, please considering providing further detail to clarify the planned physical effort of the session (exercise intensity range). In relation to the nutritional intervention, provide further detail (and source where applicable) about the five-day dietary history questionnaire, the estimation of meal sizes, the DietPerkekt dietary programme and the dietary standards for the Polish population (check the reference).
11) Suggest you make clear the primary and secondary outcome measures.
12) The two references to the Rosenberg SES need revised. Please ensure reference to the SES M is directly to the questionnaire used in this study.
13) Results – you do not mention whether you collected demographic characteristics from each study group and whether there were any differences across groups at baseline. To help future replication of studies, please provide baseline demographic information.
14) Results – suggest you include mention of flow of participants through each stage of the study: recruitment/enrolment – did anybody decline participation, or ineligible?
15) Results – please clarify if there were any missing values?
16) Discussion – line 211 – Please check use of the reference cited at the end of the first paragraph – this reference does not support your last sentence. Does it belong elsewhere?
17) Discussion – suggest you start the discussion section with reference to your own findings before generalizing.
18) Discussion – line 217 – Please clarify what you mean “…a diet with reduced energy about demand…?” and the use of all in the following “…was the most beneficial of all interventions”
19) Discussion – There are a few unsupported statements throughout the discussion, please provide reference to supporting evidence.
20) Please revise the conclusion section and clarify what you mean when you say …”the review of scientific literature…”
Author Response
Dear Reviewer,
Thank you for your favorable review of the manuscript. Any changes have been marked in red in the text. Below are the responses to the various sections of your review:
1) The notation was corrected.
2) The record has been expanded.
3) Updated.
4-7 Eligibility criteria were expanded to include required elements, including how to select and calculate minimum sample size.
8) Medical consultation has been more broadly described.
9) A repetitive passage was removed.
10) Descriptions of interventions have been revised as suggested by the Reviewer, in their current form they are clearer. Additional terms and tools used have also been clarified.
11) Descriptions of outcomes have been corrected.
12) Reference has been corrected.
13) Demographics were added.
14-15 Described in the methodology section.
16) Corrected.
17) Included as suggested.
18) Clarified unclear wording.
19) Corrected.
20) Removed questionable wording.
Greetings!
Reviewer 2 Report
The study entlitled „Global self-esteem among women 60+ with dietary and physi- 2 cal activity intervention after COVID-19 lockdown” is very interesting and well planned. In addition, such a large study groups increase the value of this study. However, minor corrections are necessary to clarify my doubts, which would certainly also appear in the readers.
My first question is why the authors emphasize the COVID pandemic and lockdown? Of course it is important, but it was not the subject of the study. We do not know if there were any changes in global self-esteem among the participants after lockdown, because the authors did not assess this. I my opinion authors should't put it in the title. The same changes could have been observed in this research group before the pandemic.
In the introduction, I miss a clear explanation of why such research was undertaken, i.e. what is already known about this? has such studies been carried out before? Why do the authors compare diet and exercise?
There is a repetition of the first paragraph in the methods (lines 84-86 = lines 96-108)
Study procedure, research tool, and interpretation:
· - For me, it is not clear if the daily level of physical activity was monitored for everyone?
· - Have you asked about possible weight changes due to the lockdown?
· - Were eventual changes after education controlled in the control group?
· - Was the energy value of the diet also assessed for the PA group?
· - Is the DietPerfekt program based on current Food Composition Tables?
· - Line 142 – „diet was reduced by 20% of total metabolism” – missing information on how total metabolism was estimated? Calculated? What pattern?
· - Line 144 – „measurements were checked every 4 weeks” - How many times have anthropometric measurements been performed for other groups? Only for the DI group we have a description that every 4 weeks.
· - Line 152 – „Polish adaptation of Łaguna” - Was the tool validated for this age group?
· - What was the initial BMI of the study participants? The authors only reported the mean value.
Results – tables
In all tables it is not clear whether the p-value (what kind of analysis is that?) is for changes after the intervention or for differences between groups? If between the groups, which ones? Were there no significant differences between the two intervention groups?
In table 2 we have an interpretation (and the description is missing in the materials and methods), while in table 3 there is no interpretation.
Have the authors assessed the relationship between body weight changes and self-esteem?
Have the authors considered comparing BMI to the values suggested for the elderly?
Discussion
The first two paragraphs are more appropriate for the introduction.
In Strengths and Limitations, the authors twice indicate that further research should include larger sample, but large sumple size was the strength of this study. I my opionion, furter study should include other study groups - inactive elderly (not participating in senior clubs or University of the Third Age classes), elderly people (over 70 years old) or elderly with low self-esteem.
Author Response
Dear Reviewer,
Thank you for your favorable review of the manuscript. Any changes have been marked in red in the text. Below are the responses to the various sections of your review:
- in the introduction, I miss a clear explanation of why such a study was undertaken, i.e. what is already known about this topic? has such research been conducted before? Why do the authors compare diet and exercise?
Re 1. The introduction has been expanded to include the required points.
- In methods there is a repetition of the first paragraph (lines 84-86 = lines 96-108).
Re 2. Removed.
- it is not clear to me whether the daily level of physical activity was monitored in everyone?
Re 3. Physical activity levels were monitored in each group, but only the PA group had the care of a fitness trainer.
- were any weight changes due to the blockade asked?
Re 4. This aspect was not studied, but we thank you for your suggestion in further research.
- were any changes after education controlled in the control group?
Re 5. Yes, at each stage. Differences between groups were compared on this basis.
- Was the energy value of the diet also evaluated in the PA group?
Re 6. No, as this aspect was reserved for the DI group.
- Is the DietPerfect program based on the current Food Composition Tables?
Re 7. Yes, calculations were also added in the Diet 6 program, which is recommended by the National Center for Nutrition Education, and the DierPerfekt program is based on the same recommendations.
- line 142 - "the diet was reduced by 20% of total metabolism." - No information on how the total metabolism was estimated? Was it calculated? What formula?
Re 8. The calculations were made in the mentioned diet programs.
- line 144 - "measurements were checked every 4 weeks". - how many times
anthropometric measurements were taken for other groups? Only for the DI group do we have a description that every 4 weeks.
Re 9. This notation has been corrected. It applied to all groups.
- line 152 - "Polish adaptation of Laguna" - Has the tool been validated for this age group?
Re 10 - Yes.
- what was the baseline BMI of the study participants? The authors provided only the mean value.
Re 11Corrected in the text.
- In all tables, it is not clear whether the p-value (what is the analysis?) refers to changes after the intervention or differences between groups? If between groups, which ones? Were there no significant differences between the two intervention groups?
Re 12 Corrected. The text describes the relationships. P-values less than 0.05 indicate a relationship between groups.
- in Table 2 we have an interpretation (and the description is missing in Materials and methods), but in Table 3 there is no interpretation.
Did the authors evaluate the relationship between weight changes and self-esteem?
Re 13 Corrected.
- did the authors consider comparing BMI to suggested values for the elderly?
Re 14. Yes.
- the first two paragraphs are more suitable for an introduction.
Re 15. Inserted into the introduction.
- In the Strengths and Limitations section, the authors twice note that further research should include a larger sample, but the large sample size was a strength of this study. In my opinion, further research should include other research groups - inactive older people (not participating in senior clubs or University of the Third Age classes), older people (over 70) or older people with low self-esteem.
Re 16 Included in the limitations of the study.
In addition, the COVID-19 aspect has been removed from the title.
Greetings!